# GSDM gene polymorphisms regulate the IgE level in asthmatic patients

**Amer Imraish** [1]*, **Tuqa Abu-Thiab**[1], **Tareq Alhindi**[1], **Malek Zihlif**[2]*

**1** Department of Biological Sciences, School of Science, The University of Jordan, Amman, Jordan,
**2** Department of Pharmacology, School of Medicine, The University of Jordan, Amman, Jordan

* a.imraish@ju.edu.jo (AI); m.zihlif@ju.edu.jo (MZ)

**Data Availability Statement:** All relevant data are within the paper and its Supporting Information files.

**Funding:** This research was funded from the deanship of academic research at the University of

## Abstract

### Background

Gasdermin A (GSDMA) and gasdermin B (GSDMB) have been associated with childhood and adult asthma in many populations including the Jordanian population. It is also known that IgE plays a crucial role in various allergic disorders, such elevated levels of total serum IgE were detected in asthma and allergic rhinitis. IgE immunoglobulin is responsible for the release of numerous inflammatory mediators, such as histamine and prostaglandins, from mast cells in asthmatic patients.

### Objective

In this study, single nucleotide polymorphisms of GSDMA (rs7212938, T/G) and GSDMB (rs7216389, T/C) in Jordanian population were investigated for their association with total IgE levels in serum of asthmatic children and adult subjects.

### Methods

The genetic polymorphism analysis for SNPs was performed using the polymerase chain reaction (PCR)/restriction fragment length polymorphism method (RFLP). Three analysis models were applied to the genotype data: co-dominant, dominant and recessive.

### Results

Our data demonstrate a significant correlation between GSDMB genetic SNP (rs7216389) and the total IgE serum level. Where one minor allele in the GSDMB gene is sufficient to induce significant changes in the IgE serum levels and plays a role in the pathogenesis of asthma in asthmatic children of the Jordanian population. Suggesting that this polymorphism might have a protective effect against asthma risk. While the presence of the GSDMB polymorphism alone might not be sufficient to associate with the high risk of developing asthma or responding to it in adults in Jordanian population.

Jordan. The funders had no role in study design, data collection and analysis, decision to publish, or preparation of the manuscript.

**Competing interests:** The authors have declared that no competing interests exist.

## Conclusion

In conclusion, the current study confirms the significant association of GSDMB genetic SNP (rs7216389) with IgE levels in asthma patients in Jordanian population, while no significant correlation of GSDMA and IgE level was found in both child and adult asthmatic patients.

## Introduction

Asthma is a complicated chronic airway constriction inflammatory disease caused by immunological abnormalities that develop both in adults and children. Asthma is an immunological abnormality characterized by a chronic airway constriction that develop both in adults and children. The hallmarks of the disease are the airway obstruction, wheezing, accumulation of IgE antibodies in response to inhaled allergens and broncho-hyperresponse [1]. Asthma is the most common chronic disease and the most common medical emergency in children [1, 2]. Despite its devastating impact, many aspects of the molecular mechanism of asthma, and hence possible therapies, remain unknown. The symptoms are usually related to obstruction, wheezing, accumulation of IgE antibodies in response to inhaled allergens, and broncho hyperresponsiveness [3, 4]. Interestingly, asthma characteristics and health outcomes appear to be age and sex-related, where it is usually atopic and associated with allergen exposure in children, displaying IgE overexpression with an excessive airway constriction and hyperresponsiveness [5, 6]. Asthma in adulthood is less likely to be associated with allergens and appears to be more frequent in women rather than men [7]. Usually, adulthood asthma is hard to be treated and shows persistent resistance to efficient conventional treatments of childhood asthma [7, 8].

Asthma is a heritable disorder that appears through subsequent generations, where 60% of the recorded asthma cases have a family history of the disease [9]. Ongoing association studies help to clarify mechanisms involved in the early onset of the disease and help to discover new therapeutic targets for this persistent disorder [7]. Recently, many genetic association and susceptibility studies that link IgE serum levels to asthma susceptible genes in childhood and adults are ongoing [10], a claim that proposes the idea of a possible genetic linkage between asthma susceptibility genes and IgE regulatory genes. Relatively, many previous genome-wide association studies show that there is an association between the Gasdermin A and Gasdermin B polymorphisms and susceptibility to adult and childhood asthma in different populations, including the Jordanian population [11]. The human GSDMA and GSDMB genes are located on chromosome 17 (17q12-21) together with other prime candidates asthma genes such as ORM1-like gene (ORMDL3) and post-GPI attachment to proteins 3 (PGAP3) [12]. In a previous study of our research group, it has been reported that genes in 17q12-21 proximal, core and distal regions are associated with asthma, the core region containing the GSDMB gene has been associated with early onset of asthma [11]. Also, it is associated with childhood asthma, and Rhinovirus (RV)—induced wheezing interaction in early life, which is linked to subsequent development of asthma. The association between GSDMB SNP (rs7216389) and susceptibility to adult and childhood asthma among Jordanians has been established in a previous study [11]. Moreover, a study that was held on Japanese also confirmed such association between rs7216389 SNP of GSDMB with asthma in children [13].

IgE immunoglobulin antibody elevated serum level has been correlated to allergic rhinitis and asthma, and it is a key for pathogenicity of allergic disorders [14]. IgE immunoglobulin is responsible for the release of many inflammatory mediators in asthma from mast cells such as

histamine and prostaglandins [15]. These inflammatory mediators promote further constricted airways by causing excessive excretion of mucus. However, in a study that evaluated whether asthma was a direct cause of allergic sensitization, they demonstrated that asthmatics were susceptible to overexpressed IgE upon exposure to common allergens in the atmosphere, where direct exposure to allergens was not a direct reason of high risk of asthma development, results suggest the presence of genetic factors manipulating IgE production and expression level in asthma [10]. Interestingly, Moffatt and colleagues had stated that there is an overlapping in a small region between genes controlling IgE levels and asthma susceptible genes [7]. One study on Amish population provided evidence for genetic linkage between IL-4 at the location 5q31.1 to the noncognate total IgE serum levels, indicating that IL-4 gene or adjacent genes control the expression of IgE in the absence of allergic antigens [16].

Taking together these data, we propose that the SNPs of GSDMA/GSDMB genes may increase susceptibility to asthma in childhood and adulthood populations through augmentation of IgE production. The aim of this study is to investigate whether there is a genetic association between gasdermin family (GSDMB/GSDMA) and serum level of IgE in children vs adult asthma patients in Jordan.

## Methods and materials

### Adult and pediatric subjects

A total of 392 subjects who visited Jordan University hospital chest and ENT clinics participated in this study; they were categorized as follow; 158 children and 234 adults (Table 1). All the participants signed an informed consent form for their approval. The consent form for children signed by their parents or carers. Depending on their history, physical examination and pulmonary imaging and tests, 46 subjects of children population and 123 subjects of the adult population were diagnosed with asthma. Remaining control subjects of both populations did not encounter any asthmatic or other allergic disorders. The study protocol was approved from the Institutional Review Board of University of Jordan [IRB# 67/2011/2012]. All patients and control subjects had signed a written informed consent. The consent form for children has been signed by their parents or carers. All procedures performed in studies involving human participants were in accordance with the ethical standards of the institutional and/or national research committee and with the 1964 Helsinki declaration and its later amendments or comparable ethical standards.

### Blood withdrawal and DNA extraction

Samples of peripheral blood of all participants were collected in EDTA tubes. Following the manufacturer's protocol of Wizard Genomic DNA Purification Kit (Purchased from Promega/USA), DNA was extracted and quantified at 260 nm and 280 wavelengths using Biorad-Red SmartSpec Plus/USA UV-Spectrophotometer, with a purity ranging from 1.7 to 2.0 being confirmed. Extracted DNA was stored at -20°C until being used for further analysis.

**Table 1. Demographic characteristics of the study population.**

|  | Sample size | % Females/ % Males | Average age±SD | Age range | Average IgE level IU/ml |
|---|---|---|---|---|---|
| Children asthma | 46 | 0.32/0.68 | 5.64±4.43 | 4 months-16 years | 172.4 |
| Children control | 112 | 0.38/0.62 | 7.58±4.82 | 1–16 years | 21.25 |
| Adult asthma | 123 | 0.72/0.28 | 44.02±14.95 | 18–84 years | 134.14 |
| Adult control | 111 | 0.46/0.54 | 38.6±14.85 | 18–76 years | 17.22 |

### PCR-RFLP polymorphism genotyping

In brief, Forward and reverse primers (GSDMA (forward:5′– GAAGGTGAAGGGAACGG
CAG-'3 and reverse: 5′–GTCACACTGGAGCGAGC CG-'3) GSDMB (forward:5′–TGTCA
CATTTCCACCAGTTCC-'3 and reverse: 5′–TGTCACATTTCCACCAGTTC C-'3)) were
used to amplify the DNA fragments and were digested using *Hpych4III* and *NsiI* restriction
enzymes for GSDMA and GSDMB products, respectively. The resultant fragments were sepa-
rated using 3% agarose gel electrophoresis and the bands were visualized using UV light.

### Measurement of Total IgE serum level

Enzyme-linked Immunosorbent Assay method was used to measure total IgE serum level.
Samples of peripheral blood of all participants were collected in EDTA tubes, and centrifuged
at 1,500 xg for 10 minutes at 4˚C, then the serum supernatant was aspirated in a clean collec-
tion tube and stored at -80˚C until being used. Following the manufacturer instructions of
Total human IgE ELISA assay kit (cat. no. RE59061; IBL International, Corp.), colour intensi-
ties were measured at 450 nm wavelength using microtiter plate reader and further statistical
analysis for measuring concentrations was performed using GraphPad Prism 7.0 software.

### Statistical analysis

GraphPad Prism 7.0 software (GraphPad Software, La Jolla, CA) was used for statistical analy-
sis of the collected data. The normality of data was determined using a Pearson normal distri-
bution curve, and the results showed that all data were normally distributed. After passing
normality test, and According to GraphPad prism software guidelines, student t-test was used
to perform comparisons, where statistical significance has been considered with $P < 0.05$
[Delacre, 2017 #150]. Results are presented as mean± S.E.M.

## Results

Gasdermin A (GSDMA) and gasdermin B (GSDMB) genes are located at chromosome 17, and
variants of these genes are confirmed to increase susceptibility to asthma phenotypes in chil-
dren. In this study, single nucleotide polymorphisms of GSDMA (rs7212938) and GSDMB
(rs7216389) in Jordanian population were investigated for their association with total IgE lev-
els in serum of asthmatic children and adult subjects. A total of 158 children and 234 adults
were tested and genotyped for the GSDMA and GSDMB variants, and genotypes were catego-
rized into co-dominant, dominant, and recessive genotypic models. Major and minor alleles
for GSDMA gene are G and T respectively, while for GSDMB gene are T and C, respectively.

To clarify the correlation among asthmatics, children subjects were categorized into asth-
matics and controls (Table 2). Among 158 tested children, 112 were healthy controls and 46
were diagnosed with asthma. For GSDMA SNP (rs7212938), our statistical analysis results
indicate that this SNP is not significantly associated with the IgE serum level among the three
genotyping models for both control and asthmatic groups in children population. Interest-
ingly, asthmatic children showed a noticeable drop in the IgE serum level in the GT heterozy-
gous and TT minor pure genotypes (128.8 and 114.8 IU/ml, respectively), which are around
half of that for GG major pure genotype (308.7 IU/ml), but our correlation results indicated
that these differences did not reach the significant level and P-values were much higher than
0.05.

On the contrary, GSDMB single nucleotide polymorphism (rs7216389) did have a signifi-
cant correlation with IgE serum levels in both control and asthmatic children. In the control
subjects, comparison between IgE levels of major homozygous trait and heterozygous was

**Table 2. Comparison between IgE levels of children asthma patients and control group according to GSDMA and GSDMB.**

| | Model | Genotype | Control (n = 112) IgE levels: IU/ml (n) | IgE Mean comparison | P-value | Asthma (n = 46) IgE levels: IU/ml (n) | IgE Mean comparison | P-value |
|---|---|---|---|---|---|---|---|---|
| GSDMA | Co-dominant | GG | 19.9±0.4 (39) | GG vs GT | 0.2005 | 308.7±6.6 (12) | GG vs GT | 0.7032 |
| | | GT | 22.05±0.7 (56) | GG vs TT | 0.7031 | 128.8±2.9 (23) | GG vs TT | 0.9278 |
| | | TT | 13.3±0.8 (17) | GT vs TT | 0.5791 | 114.8±0.8 (11) | GT vs TT | 0.7806 |
| | Dominant | GG | 19.9±0.4 (39) | | 0.2427 | 308.7±6.6 (12) | | 0.7500 |
| | | GT+TT | 20.0±0.7 (73) | | | 124.3±2.0 (34) | | |
| | Recessive | GG+GT | 21.2±0.6 (95) | | 0.8891 | 190.5±5.1 (35) | | 0.8892 |
| | | TT | 13.3±0.8 (17) | | | 114.8±0.8 (11) | | |
| | Model | Genotype | Control (n = 112) IgE levels: IU/ml (n) | IgE Mean comparison | P-value | Asthma (n = 46) IgE levels: IU/ml (n) | IgE Mean comparison | P-value |
| GSDMB | Co-dominant | TT | 20.0±0.2 (22) | TT vs TC | 0.1544 | 865.0±9.1 (4) | TT vs TC | 0.0308 |
| | | TC | 16.8±0.8 (48) | TT vs CC | 0.0299 | 188.3±5.8 (17) | TT vs CC | 0.0080 |
| | | CC | 23.5±0.7 (42) | TC vs CC | 0.3315 | 111.1±1.7 (25) | TC vs CC | 0.9179 |
| | Dominant | TT | 20.0±0.2 (22) | | 0.0428 | 865.0±9.1 (4) | | 0.0059 |
| | | TC+CC | 19.9±0.8 (90) | | | 142.4±3.2 (42) | | |
| | Recessive | TT+TC | 17.8±0.6 (70) | | 0.1086 | 317.2±6.5 (21) | | 0.3206 |
| | | CC | 23.5±0.7 (42) | | | 111.1±1.7 (25) | | |

insignificant with P = 0.1544, and IgE level was slightly dropped from 20 IU/ml for TT into 16.8 IU/ml for TC, where comparing major TT trait with CC minor trait shown to be significant with P-value = 0.0299 with a slight increase in the IgE serum level for the minor homozygous genotype (23.5 IU/ml). No significant change was observed upon comparing heterozygous and minor pure genotypes (P- values = 0.3315). These data suggest that the presence of both mutated alleles in the co-dominant model was needed to induce a significant change in IgE serum levels in control children subjects. Furthermore, a massive drop in the IgE serum level was observed among TT major, TC heterozygous and CC minor genotypes of the co-dominant model in asthmatic children (865 IU/ml, 188.3 IU/ml and 111.1 IU/ml, respectively). Comparison data indicate that the association between GSDMB SNP and IgE serum level is significant in asthmatic children, where the presence of one minor allele in the heterozygous genotype (TC) significantly lowered IgE serum level about 4.5 times the serum levels of normal major genotype (TT) with P-value = 0.0308. Similarly, a further significant reduction (about 8 times) in the IgE serum level was achieved when comparing major pure with minor pure genotypes in the co-dominant model (P-value = 0.008). These data suggest that one minor allele in the GSDMB gene is sufficient to induce significant changes in the IgE serum levels and plays a role in the pathogenesis of asthma in asthmatic children of the Jordanian population. Our data also shows that GSDMB SNP is significantly correlated with the IgE serum levels in the dominant model of both control and asthmatic children subjects (P-values = 0.0428 and 0.0059, respectively), but the insignificant association was found for the recessive model of the same groups of children population.

To investigate the effect of SNPs in adulthood and to figure out whether they have any correlation with IgE expression levels, an association analysis was also performed on adults. A detailed correlation analysis between GSDMA and GSDMB tested SNPs with IgE serum level was performed (Table 3), where adult subjects were categorized into healthy controls and asthmatics. As shown in Table 3, rs7212938 SNP of GSDMA has no association with IgE expression in adults for all genotyping models, even though IgE levels were noticeably higher in asthmatic subjects in comparison to healthy subjects. Same IgE level pattern was observed for

**Table 3. Comparison between IgE levels of adult asthma patients and control group according to GSDMA and GSDMB.**

| | Model | Genotype | Control (n = 111) IgE levels: IU/ml (n) | IgE Mean comparison | p-value | Asthma (n = 123) IgE levels: IU/ml (n) | IgE Mean comparison | P-value |
|---|---|---|---|---|---|---|---|---|
| GSDMA | Co-dominant | GG | 16.5±3.1 (47) | GG vs GT | 0.4027 | 119.2±2.2 (48) | GG vs GT | 0.4140 |
| | | GT | 16.6±3.0 (44) | GG vs TT | 0.1744 | 152.7±6.2 (54) | GG vs TT | 0.4415 |
| | | TT | 20.1±5.4 (20) | GT vs TT | 0.4815 | 120.5±4.8 (21) | GT vs TT | 0.1482 |
| | Dominant | GG GT +TT | 16.5±3.1 (47) | | 0.2358 | 119.2±2.2 (48) | | 0.8035 |
| | | | 17.5±4.1 (64) | | | 143.7±5.8 (75) | | |
| | Recessive | GG+GT TT | 16.6±3.0 (91) | | 0.2397 | 136.9±5.2(102) | | 0.2074 |
| | | | 20.1±5.4 (20) | | | 120.5±4.8 (21) | | |
| | Model | Genotype | Control (n = 111) IgE levels: IU/ml (n) | IgE Mean comparison | P-value | Asthma (n = 123) IgE levels: IU/ml (n) | IgE Mean comparison | P-value |
| GSDMB | Co-dominant | TT | 17.92±0.8 (18) | TT vs TC | 0.9272 | 146.8±4.8 (22) | TT vs TC | 0.0392 |
| | | TC | 14.3±0.4 (58) | TT vs CC | 0.2561 | 93.8±2.6 (56) | TT vs CC | 0.4808 |
| | | CC | 21.6±0.9 (35) | TC vs CC | 0.1290 | 178.0±7.6 (45) | TC vs CC | 0.0997 |
| | Dominant | TT TC +CC | 17.92±0.8(18) | | 0.5700 | 146.8±4.8(22) | | 0.1097 |
| | | | 17.1± 5.8 (93) | | | 131.3±5.8(101) | | |
| | Recessive | TT+TC CC | 15.2±0.5 (76) | | 0.1017 | 108.8±3.0(78) | | 0.4250 |
| | | | 21.6±0.9 (35) | | | 178.0±7.6(45) | | |

GSDMB SNP in adults, where It was much higher in asthmatics compared to healthy controls. Interestingly, a significant association was found when comparing major homozygous genotype with heterozygous genotype in the co-dominant model, where the presence of one minor allele was enough to induce a significant drop in the IgE serum level in the adult asthmatic group. Surprisingly, IgE serum level increased again in the minor homozygous genotype of GSDMB and scored the highest IgE serum level among all other genotypes in asthmatic groups. The same pattern was also observed in the control group where heterozygous genotype scored the lowest IgE level, and minor homozygous genotype scored the highest IgE serum levels. No significant correlation between GSDMB tested SNP and IgE serum level was found for the dominant and recessive models in both Asthmatic and control adult subjects.

Table 4 shows the results of the one-way ANOVA statistical analysis between the tested SNPs of both genes and the clinical respiratory F-ratio parameter [forced expiratory volume (FEV1) and forced vital capacity (FVC)]; which is the ratio between the FEV1 (Forced expiratory volume) and FVC (Forced vital capacity), in both the control and asthmatic subjects. Moreover, in Table 4, 2-way ANOVA multiple comparison analysis for the association of the SNPs with both serum IgE levels and F-ratio are also shown. Regarding GSDMA tested SNP (rs7212938, T/G) in asthmatic patients, the associations between the SNP and the F-ratio was significant between the major homozygous (GG) and heterozygous (GT) genotypes ($P = 0.0457$), and between heterozygous (GT) and minor homozygous (TT) genotypes ($P = 0.0497$). No significant association was found between the SNP and the F-ratio among the control subjects, with P values $> 0.05$. In the 2-way ANOVA analysis of GSDMA tested SNP (rs7212938) with both serum IgE levels and F-ratio, significant association was found in both control and asthmatic groups, with p-values $< 0.0001$ for controls and $P = 0.0345$ for asthmatics.

For GSDMB gene, one-way ANOVA statistical analysis of GSDMB SNP (rs7216389, T/C) and the F-ration did not show any significant association among both asthmatic and control groups, with p-values $> 0.05$. Results of 2-way ANOVA analysis between GSDMB SNP, F-ration and total IgE serum level were significant for control subjects only with p-value $<0.0001$.

**Table 4. Association of the lung function for the adult asthmatic patients with GSDMA and GSDMB and IgE levels.**

|  | SNP | Asthma F-ratio | Asthma (n = 123) IgE levels: IU/ml (n) | Association between SNP and F-ratio One-way ANOVA P-value | | Association between SNP, F-ratio and IgE level 2-way ANOVA P-value |
|---|---|---|---|---|---|---|
| GSDMA | GG | 77.4±1.3 | 119.2±2.2 (48) | GG vs GT | 0.0457 | 0.0345 |
|  | GT | 75.6±1.4 | 152.7±6.2 (54) | GG vs TT | 0.8760 |  |
|  | TT | 77.8±2.3 | 120.5±4.8 (21) | GT vs TT | 0.0497 |  |
|  | SNP | Control (n = 95) F-ratio (n) | Control (n = 111) IgE levels: IU/ml (n) | Association between SNP and F-ratio One-way ANOVA P-value | | Association between SNP, F-ratio and IgE level 2-way ANOVA P-value |
| GSDMA | GG | 101.8±0.9 | 16.5±3.1 (47) | GG vs GT | 0.9088 | <0.0001 |
|  | GT | 101.2±1.08 | 16.6±3.0 (44) | GG vs TT | 0.8474 |  |
|  | TT | 101.7±1.32 | 20.1±5.4 (20) | GT vs TT | 0.6899 |  |
|  | SNP | Asthma(n = 123) F-ratio | Asthma (n = 123) IgE levels: IU/ml (n) | Association between genotype and F-ratio One-way ANOVA P-value | | Association between SNP, F-ratio and IgE level 2-way ANOVA P-value |
| GSDMB | TT | 77.4±1.3 | 146.8±4.8 (22) | TT vs TC | 0.7937 | 0.2136 |
|  | TC | 75.6±1.4 | 93.8±2.6 (56) | TT vs CC | 0.4532 |  |
|  | CC | 77.8±2.3 | 178.0±7.6 (45) | TC vs CC | 0.7199 |  |
|  | SNP | Control(n = 111) F-ratio | Control (n = 111) IgE levels: IU/ml (n) | Association between genotype and F-ratio One-way ANOVA P-value | | Association between SNP, F-ratio and IgE level 2-way ANOVA P-value |
| GSDMB | TT | 102.6±1.01 | 17.92±0.8 (18) | TT vs TC | 0.8609 | <0.0001 |
|  | TC | 102.2±0.84 | 14.3±0.4 (58) | TT vs CC | 0.7824 |  |
|  | CC | 99.8±1.2 | 21.6±0.9 (35) | TC vs CC | 0.2678 |  |

## Discussion

The association between GSDMA (rs7212938, T/G) and GSDMB (rs7216389, T/C) at locus 17q12-21 and level of IgE immunoglobulin antibody in child and adult Jordanian asthma patients has been tested. Our data illustrate a strong association between GSDMB and IgE level in serum of asthmatic children. rs7216389 SNP of GSDMB gene significantly reduced expression of IgE in asthmatic children serum, while the presence of one minor allele was enough to induce these changes. It is noteworthy to highlight the difference in IgE serum levels between control and asthmatic groups of both children and adult populations, where IgE levels are noticeably elevated in all genotypes of asthmatics in comparison to those of control subjects. Interestingly, GSDMB SNP (rs7216389) in the present study showed a protective effect in asthmatic children, and the minor homozygous genotype reduced IgE levels significantly. This confirms what has been reported that GSDMB CC homozygous minor genotype had a protective effect against an IgE related diseases such as asthma and allergic rhinitis risk [11].

In a genome-wide association study, it was reported that 17q21 loci which include ORMDL3/GSDMB genes is highly correlated with childhood asthma, but unlike our results, GSDMB is not correlated to total IgE serum levels [7]. This is explained by their different tested SNP, where they tested GSDMB SNP (rs2305480), while in our study we tested (rs7216389) SNP [7], though different SNPs may reflect different consequences and functions.

In the adult population, a significant association of GSDMB SNP (rs7216389, T/C) and IgE level in asthmatic patients was found only in the co-dominant model, when comparing major homozygous to heterozygous genotype (TT vs TC, p-value 0.0392), which indicate a significant drop in the level of IgE with TC genotypes and again elevation in the minor homozygous genotypes. A possible reason for this fluctuating IgE levels among genotypes is the low frequency of the major homozygous genotype, where more candidates needed to be involved to clarify the effect of our tested SNP on IgE levels, in addition to the involvement of other environmental

or genetic factors that might has an impact on the release of IgE. These data suggest that the presence of the GSDMB polymorphism alone might not be sufficient to associate with the high risk of developing asthma or responding to it in adults in the Jordanian population, results that are in line to our previous association study between asthma and GSDMB SNP (rs7216389, T/C) [11].

In a study similar to ours and was conducted on Japanese adults, they also demonstrated a significant association between rs7216389 GSDMB SNP and total IgE serum levels, that causes an elevation in IgE levels in a way independent of asthma [17]. They suggested that ORMLD3/ GSDMB region could play a role in susceptibility and high-risk development of asthma, by increasing total IgE production which was enhanced by external environmental interplays such as tobacco smoke or viral infections, but not allergic sensitization or allergic rhinitis [17]. One suggested mechanism was the activation of innate type-2 immunity and subsequent air-way inflammatory response after being exposed to the external interplaying factors [17, 18]. T helper cells are a key regulator during asthma pathogenesis; they regulate the secretion of many cytokines, including IL-13, IL-5 and IL-4 and involved in innate type 2 immunity [19]. It was reported previously that there is an overlapping between asthma susceptible genes at loci 17q12-21 and genes controlling the expression of IgE, including IL-4 gene [16]. Hence, we suggest a possible linkage between GSDMB gene and IL-4 gene, and the tested rs7216389 SNP of GSDMB may affect the expression of IL-4 cytokine. It is recommended to investigate further the association between GSDMB and secretion of different cytokine including IL-4, IL-13 and IL-5, in order to illustrate the mechanism how GSDMB tested SNP manipulate IgE serum levels in asthmatic children. Importantly, the results indicate that GSDMB polymorphism is not associated with the lung function in both the adults and children's asthmatic patients. This suggests that the predominant genetic effect of the GSDMB polymorphism is on allergy and not asthma per se. Collectively, these data support our suggestion about the significant correlation between GSDMB and IgE serum level in asthmatics, but in a way independent on asthma status, in addition to the involvement of other factors that interfere with this association.

In case of GSDMA, no significant correlation was found to IgE level both in child and adult asthmatic groups, even though it was reported in the previous study [11] that there was a significant correlation in GSDMA recessive model (GG+GT, TT) but only after multiple variation correction. Interestingly, our results indicate that is GSDMA polymorphism is significantly associated with the respiratory function of the asthmatic group and not with the control subject. Such association indicates that GSDMA polymorphism may play as a risk factor in the non-IgE dependent asthma. The association of GSDMA and GSDMB with asthma in children and adults was and still a debated topic, Moffatt, Gut [7] denied an association of GSDMA/GSDMB with adult asthma. However, the association is supported by other studies. Marinho, Custovic [20] supported the association of both genes to adult asthma, while Qiu, Zhao [21] and Wan, Shrine [22] only supported the relationship in GSDMB.

## Conclusion

In conclusion, the finding of this study confirms the association between the GSDMB and the IgE levels in child and to a lesser extent, adult asthmatic patient in Jordanian population. GSDMB SNP may increase susceptibility to asthma development in childhood, but in a way independent of IgE release. In contrast, no significant association was found between GSDMA and IgE levels in both child and adult asthmatic patients. The reported association between GSDMB and child asthma may point to the heterogeneity in the pathophysiology of asthma in children where other factors might influence the IgE serum level. It is recommended to evaluate further the involvement of interfering environmental factors on the association between

GSDMB SNP and IgE serum levels. Furthermore, the mechanism that allows GSDMB SNP to be high-risk factor for developing asthma exclusively in childhood but not adulthood should be further investigated.

## Experimental limitations

Our results reflected the patients and controls who are included in the study; although our sample size is considered convenient, increasing the sample size could alter the patient/control ratio of each SNP and provide more accurate results.

## Supporting information

**S1 File.**
(XLSX)

## Acknowledgments

All this work was conducted at the University of Jordan.

## Author Contributions

**Conceptualization:** Amer Imraish, Malek Zihlif.

**Data curation:** Amer Imraish.

**Formal analysis:** Amer Imraish, Malek Zihlif.

**Funding acquisition:** Amer Imraish.

**Investigation:** Amer Imraish, Tareq Alhindi, Malek Zihlif.

**Methodology:** Amer Imraish, Tuqa Abu-Thiab, Malek Zihlif.

**Project administration:** Amer Imraish, Tareq Alhindi.

**Resources:** Amer Imraish.

**Software:** Amer Imraish.

**Supervision:** Amer Imraish, Malek Zihlif.

**Validation:** Amer Imraish.

**Visualization:** Amer Imraish.

**Writing – original draft:** Amer Imraish, Tuqa Abu-Thiab, Tareq Alhindi, Malek Zihlif.

**Writing – review & editing:** Amer Imraish.

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
