## [Decision Letter · Decision Letter 0]

18 Mar 2022

PONE-D-21-33691GSDM gene polymorphisms regulate the IgE level in asthmatic patientsPLOS ONE

Dear Dr. Imraish,

Thank you for submitting your manuscript to PLOS ONE. After careful consideration, we feel that it has merit but does not fully meet PLOS ONE’s publication criteria as it currently stands. Therefore, we invite you to submit a revised version of the manuscript that addresses the points raised during the review process.

We look forward to receiving your revised manuscript.

Kind regards,

Maria Maddalena Sirufo

Academic Editor

PLOS ONE

Journal Requirements:

2. We note that you have stated that you will provide repository information for your data at acceptance. Should your manuscript be accepted for publication, we will hold it until you provide the relevant accession numbers or DOIs necessary to access your data. If you wish to make changes to your Data Availability statement, please describe these changes in your cover letter and we will update your Data Availability statement to reflect the information you provide."

3. Please amend the manuscript submission data (via Edit Submission) to include author Tareq Alhindi.

4. Please include your tables as part of your main manuscript and remove the individual files. Please note that supplementary tables (should remain/ be uploaded) as separate "supporting information" files".

Additional Editor Comments:

it's a very interesting job.

it would be useful to know if you have measured the specific IGEs and if there were differences between the different allergies.

add the ranges of the IGE

Reviewers' comments:

Reviewer's Responses to Questions

**Comments to the Author**

1. Is the manuscript technically sound, and do the data support the conclusions?

Reviewer #1: Yes

Reviewer #2: Yes

2. Has the statistical analysis been performed appropriately and rigorously? 

Reviewer #1: No

Reviewer #2: Yes

3. Have the authors made all data underlying the findings in their manuscript fully available?

Reviewer #1: Yes

Reviewer #2: Yes

4. Is the manuscript presented in an intelligible fashion and written in standard English?

Reviewer #1: Yes

Reviewer #2: Yes

5. Review Comments to the Author

Reviewer #1: This study investigated the association of SNP of GSDMA (rs7212938, T/G) and GSDMB (rs7216389, T/C) in Jordanian population with total IgE levels in serum of asthmatic children and adult subjects. It was found that GSDMB SNP increased susceptibility to asthma development in childhood in a way independent of IgE release, and no significant association was found between GSDMA and IgE levels in both child and adult asthmatic patients. The topic is useful and the conclusion is reliable. However, a minor modify could be completed.

1. In the method section

- Please state the age range of the patients and controls.

-Please state the cat no. of the used ELISA kit

2. in the results section

-Gasdermin A (GSDMA) and gasdermin B (GSDMB) genes are located at chromosome 17, and variants of these genes are confirmed to increase susceptibility to asthma phenotypes in children (please add reference)

- Please state which clinical respiratory parameters were measured.

Reviewer #2: Based on an association between the Gasdermin A and Gasdermin B polymorphisms and susceptibility to adult and childhood asthma in the Jordanian population, the association between GSDMA (rs7212938, T/G) and GSDMB (rs7216389, T/C) at locus 17q12-21 and level of IgE immunoglobulin antibody in child and adult Jordanian asthma patients has been tested.

This study confirmed that there is a significant association between GSDMB genetic SNP (rs721638) and IgE levels in asthma patients in Jordanian population. Overall, the results are meaningful. Minor concern is below.

1. The author need to add the forward and the reverse primers sequences

2. The author need to state the name of normality test that have been used

3. When did the Peripheral blood specimens were collected from patients? at the acute attack or stable stage of asthma attack？

6. PLOS authors have the option to publish the peer review history of their article (what does this mean?). If published, this will include your full peer review and any attached files.

Reviewer #1: No

Reviewer #2: No

---

## [Author Response · Author response to Decision Letter 0]

30 May 2022

GSDM gene polymorphisms regulate the IgE level in asthmatic patients

reference: PONE-D-21-33691

Dear Editor,

Accompanying this letter are the revised version of the manuscript with all changes needed, and response to reviewers’ comments. We would like to thank the respected reviewers again for taking the time and effort to read the revised manuscript, make some corrections, and for their comments and suggestions which will definitely improve the quality of the paper. We have addressed most of the points, comments, and suggestions made by the reviewers and hope the manuscript should come to a happy ending.

We thank you for your consideration and look forward to hearing from you soon.

 reference: PONE-D-21-33691

Title: GSDM gene polymorphisms regulate the IgE level in asthmatic patients

PLOS One Reports: Amer Imraish, Tuqa Abu-Thiab, Tareq Alhindi, Malek Zihlif

Reviewers' comments:

Reviewer 1

This study investigated the association of SNP of GSDMA (rs7212938, T/G) and GSDMB (rs7216389, T/C) in Jordanian population with total IgE levels in serum of asthmatic children and adult subjects. It was found that GSDMB SNP increased susceptibility to asthma development in childhood in a way independent of IgE release, and no significant association was found between GSDMA and IgE levels in both child and adult asthmatic patients. The topic is useful and the conclusion is reliable. However, a minor modify could be completed.

1. In the method section

- Please state the age range of the patients and controls.

The age range was added to the Table 1.

-Please state the cat no. of the used ELISA kit

The following was added to the revised manuscript in the method part 

cat. no. RE59061; IBL International, Corp.

2. in the results section

-Gasdermin A (GSDMA) and gasdermin B (GSDMB) genes are located at chromosome 17, and variants of these genes are confirmed to increase susceptibility to asthma phenotypes in children (please add reference)

The reference has been added.

- Please state which clinical respiratory parameters were measured.

the following was added to the results section:

[forced expiratory volume (FEV1) and forced vital capacity (FVC)].

Reviewer #2: 

Based on an association between the Gasdermin A and Gasdermin B polymorphisms and susceptibility to adult and childhood asthma in the Jordanian population, the association between GSDMA (rs7212938, T/G) and GSDMB (rs7216389, T/C) at locus 17q12-21 and level of IgE immunoglobulin antibody in child and adult Jordanian asthma patients has been tested.

This study confirmed that there is a significant association between GSDMB genetic SNP (rs721638) and IgE levels in asthma patients in Jordanian population. Overall, the results are meaningful. Minor concern is below.

1. The author need to add the forward and the reverse primers sequences

The primers sequences were added to the revised manuscript in the method part

2. The author need to state the name of normality test that have been used

The following was added to the method part in the revised manuscript 

The normality of data was determined using a Pearson normal distribution curve, and the results showed that all data were normally distributed.

3. When did the Peripheral blood specimens were collected from patients? at the acute attack or stable stage of asthma attack?

Thanks much for your interesting comments. For the first point, the blood samples were collected in stable stage. At least not under exacerbation that needed oral cortisone.

---

## [Decision Letter · Decision Letter 1]

18 Jul 2022

PONE-D-21-33691R1GSDM gene polymorphisms regulate the IgE level in asthmatic patientsPLOS ONE

Dear Dr. Imraish,

Thank you for submitting your manuscript to PLOS ONE. After careful consideration, we feel that it has merit but does not fully meet PLOS ONE’s publication criteria as it currently stands. Therefore, we invite you to submit a revised version of the manuscript that addresses the points raised during the review process.

We look forward to receiving your revised manuscript.

Kind regards,

Dong Keon Yon, MD, FACAAI

Academic Editor

PLOS ONE

Additional Editor Comments:

I read it with great interest, but the author have to address my critical comments. For this paper to be published in the Plos One, an overall revision of the manuscript must be made.

#1. Please describe strengths and limitations section.

#2. Funding: With what research funds did the author conduct this analysis? Was it carried out at your own expense?

#3. Identify the exact inclusion criteria for 392 participants.

#4. What is the definition of asthma?

#5. Describe the IgE sampling in the method.

#6. Describe the IgE mean level in the Table 1.

#7. Did the author only conduct baseline surveys on age, IgE level, and asthma? This level of investigation cannot provide a faithful correction to the selection bias.

#8. In statistical section, the authors have to mention the statistical guideline such as https://doi.org/10.54724/lc.2022.e3

#9. I cannot believe that 300 people were not excluded from the study. Usually, 10% exclusion is made in large-scale international studies.

I cannot guarantee the publication approval of this paper unless there is an appropriate answer to my opinion.

Thank you.

Reviewers' comments:

Reviewer's Responses to Questions

**Comments to the Author**

1. If the authors have adequately addressed your comments raised in a previous round of review and you feel that this manuscript is now acceptable for publication, you may indicate that here to bypass the “Comments to the Author” section, enter your conflict of interest statement in the “Confidential to Editor” section, and submit your "Accept" recommendation.

Reviewer #1: All comments have been addressed

2. Is the manuscript technically sound, and do the data support the conclusions?

Reviewer #1: Yes

3. Has the statistical analysis been performed appropriately and rigorously? 

Reviewer #1: Yes

4. Have the authors made all data underlying the findings in their manuscript fully available?

Reviewer #1: Yes

5. Is the manuscript presented in an intelligible fashion and written in standard English?

Reviewer #1: Yes

6. Review Comments to the Author

Reviewer #1: (No Response)

7. PLOS authors have the option to publish the peer review history of their article (what does this mean?). If published, this will include your full peer review and any attached files.

Reviewer #1: **Yes: **Wajdy Al-Awaida

Department of Biology and Biotechnology, American University of Madaba, Madaba, Jordan

---

## [Decision Letter · Decision Letter 2]

8 Sep 2022

GSDM gene polymorphisms regulate the IgE level in asthmatic patients

PONE-D-21-33691R2

Dear Dr. Imraish,

We’re pleased to inform you that your manuscript has been judged scientifically suitable for publication and will be formally accepted for publication once it meets all outstanding technical requirements.

Kind regards,

Dong Keon Yon, MD, FACAAI

Academic Editor

PLOS ONE

Additional Editor Comments (optional):

This is an excellent paper.

Reviewers' comments:

Reviewer's Responses to Questions

**Comments to the Author**

1. If the authors have adequately addressed your comments raised in a previous round of review and you feel that this manuscript is now acceptable for publication, you may indicate that here to bypass the “Comments to the Author” section, enter your conflict of interest statement in the “Confidential to Editor” section, and submit your "Accept" recommendation.

Reviewer #1: All comments have been addressed

2. Is the manuscript technically sound, and do the data support the conclusions?

Reviewer #1: Yes

3. Has the statistical analysis been performed appropriately and rigorously? 

Reviewer #1: Yes

4. Have the authors made all data underlying the findings in their manuscript fully available?

Reviewer #1: Yes

5. Is the manuscript presented in an intelligible fashion and written in standard English?

Reviewer #1: Yes

6. Review Comments to the Author

Reviewer #1: (No Response)

7. PLOS authors have the option to publish the peer review history of their article (what does this mean?). If published, this will include your full peer review and any attached files.

Reviewer #1: **Yes: **Wajdy Awaida

---

## [Editor Report · Acceptance letter]

27 Sep 2022

PONE-D-21-33691R2 

GSDM gene polymorphisms regulate the IgE level in asthmatic patients 

Dear Dr. Imraish:

I'm pleased to inform you that your manuscript has been deemed suitable for publication in PLOS ONE. Congratulations! Your manuscript is now with our production department. 

Kind regards, 

on behalf of

Dr. Dong Keon Yon 

Academic Editor

PLOS ONE